# *Betholletia excelsa* Fruit: Unveiling Toughening Mechanisms and Biomimetic Potential for Advanced Materials

**DOI:** 10.3390/biomimetics8070509

**Published:** 2023-10-26

**Authors:** Marilia Sonego, Anneke Morgenthal, Claudia Fleck, Luiz Antonio Pessan

**Affiliations:** 1Graduate Program in Materials Science and Engineering (PPGCEM), Federal University of São Carlos (UFSCar), São Carlos 13565-905, SP, Brazil; pessan@ufscar.br; 2Institute of Mechanical Engineering, Federal University of Itajubá (UNIFEI), Itajubá 37500-903, MG, Brazil; 3Materials Science and Engineering, Technische Universität Berlin, 10623 Berlin, Germanyclaudia.fleck@tu-berlin.de (C.F.); 4Department of Materials Engineering, Federal University of São Carlos, São Carlos 13565-905, SP, Brazil

**Keywords:** biomimetic, Brazil nut mesocarp, *Bertholletia excelsa*, fracture analysis

## Abstract

Dry fruits and nutshells are biological capsules of outstanding toughness and strength with biomimetic potential to boost fiber-reinforced composites and protective structures. The strategies behind the *Betholletia excelsa* fruit mechanical performance were investigated with C-ring and compression tests. This last test was monitored with shearography and simulated with a finite element model. Microtomography and digital and scanning electron microscopy evaluated crack development. The fruit geometry, the preferential orientation of fibers involved in foam-like sclereid cells, promoted anisotropic properties but efficient energy dissipating mechanisms in different directions. For instance, the mesocarp cut parallel to its latitudinal section sustained higher forces (26.0 ± 2.8 kN) and showed higher deformation and slower crack propagation. The main toughening mechanisms are fiber deflection and fiber bridging and pullout, observed when fiber bundles are orthogonal to the crack path. Additionally, the debonding of fiber bundles oriented parallel to the crack path and intercellular cracks through sclereid and fiber cells created a tortuous path.

## 1. Introduction

Wood, bamboo, and fiber cells are well-known examples of how plants can inspire new technical materials [1,2,3,4]. For instance, fiber-reinforced composites can be greatly improved by strategies found in the hierarchical architecture of plants, especially considering the cellulose reinforcement of cell walls, interfaces, and cell orientations [5,6,7,8]. However, for that, the microstructure and biomechanics of different plant structures must be investigated. As recent studies of macadamia [9,10,11], babassu [6], walnut [12], and coconut [13] have shown, the fruit walls of nuts and drupes can inspire new puncture- and impact-resistant materials [14,15]. These biological structures are protective capsules with remarkably specific toughness and strength that can inspire improvements in structural composites, pressure vessels, and protective components for commodities or persons [12,16].

The fruit of *Bertholletia excelsa*, of which the edible seeds are colloquially known as “Brazil nuts”, has shown a mechanical performance comparable to or even better than many fruit walls [17]; therefore, it is also an excellent source of bioinspiration. This plant has a dry and woody fruit with a round shape of 10–12 cm diameter, and an outer wall with a thickness of 1–2 cm that protects seeds against predators and falls from trees as high as 50 m [18]. The woody fruit is difficult to open. Only animals with sharp teeth and a strong biting force can consume them [19]. Every fruit, also known as pericarp, has three layers called the endocarp, mesocarp, and endocarp. In contrast with many other nutshells, *Bertholletia excelsa* pericarp has no suture line, making it even more challenging to open. However, this fruit has a peduncle (stem) and an opercular opening at opposite ends, as shown in Figure 1a,b. The opercular opening evolved to have a smaller diameter than the seeds and hold them safely inside the fruit.

The mesocarp is the thickest layer of Brazil nut fruit and the primary protector of the seeds [17]. It is mainly composed of sclereids and fiber cells (Figure 1c) [20]. The spherical or elliptical sclereids and the elongated fibers have thick and highly lignified cell walls, and consequently, they are hollow after cell death during fruit maturation [21]. As shown in our morphological study of the Brazil nut fruit [20], the mesocarp has a sandwich structure of three layers, distinguished depending on the preferential orientation of the fibers (Figure 1d). In the most internal (“I”) and external (“III”) layers, groups of fibers, that is, fiber bundles, are oriented with their long axis parallel to the direction from the peduncle to the opercular opening (“longitudinal” orientation—Figure 1a). In layer II, the thickest one, there is a high density of fibers that are latitudinally oriented, that is, orthogonal to the fibers in layer I (“latitudinal” orientation—Figure 1b). Layer I has ridges running from the peduncle to the opercular opening; these ridges are composed of highly oriented fibers, which are highlighted in Figure 1b. The *Bertholletia excelsa* mesocarp shows great resemblance to some fiber reinforcements composites. For instance, it has fibers in different orientations forming a sandwich structure while involved in a foam-like matrix with very different properties. Therefore, the understanding of how this biological material achieves outstanding mechanical performance can provide useful insights to optimize mechanical properties while reducing the density of fiber-reinforced composites. Such a biological capsule can inspire improvements on pipes/tubes, pressure vessels, tanks, masts, missile cases, rocket motor cases, aircraft fuselages, and protective components for commodities or persons.

Another interesting feature of mesocarp morphology is the presence of voids everywhere within the structure, here named structural voids [20]. Some of these voids have a tubular shape and are remnants of the plant’s vascular system, which are necessary for water and nutrient distribution to the living cells [20,22]. The channel-like voids were also identified in coconut endocarp [13] and in macadamia seed shells [9,11], where they act as toughening mechanisms by trapping and deflecting advancing cracks. In macadamia, such voids have also been identified as stress raisers, depending on size, orientation, and distribution [9]. Other mesocarp voids in Brazil nuts appear as cracked surfaces, possibly formed by hydrothermal deformations of the weak boundaries between sclereids and fiber bundles. A recent study has shown that both kinds of mesocarp voids can have the same dual effect, acting either as toughening mechanism or as stress raisers, depending on their size, orientation, and distribution [23].

This study aims to characterize the macro-mechanical performance of Brazil nut mesocarps by compression and tensile C-ring tests and compression tests. Additionally, fracture analysis, finite element modeling (FEM) and crack monitoring with scanning electron microscopy (SEM), shearography, and microtomography (microCT) aims a correlation between mechanical properties, fruit geometry, and microstructure. The understanding of how sclereids, fiber bundles, voids, and mesocarp geometry affect the mechanical performance of this biological capsule can lead to useful insights to bioinspired materials.

## 2. Materials and Methods

### 2.1. Materials

Fruits of *Bertholletia excelsa* were bought from a local producer from Jaboticabal, SP, and Breves, PA, in Brazil. Each fruit was cleaned and cut with a band saw to sections shown in Figure 1a,b. The remaining exocarp, endocarp, tegument, and seeds were discarded. Only the mesocarp structure was tested.

### 2.2. C-Ring

Compression and tensile C-ring tests were performed to evaluate the fracture strength and elastic modulus on Zwick universal testing machine, equipped with Inspect retrofit 1475 by Hegewald & Peschke Meß- und Prüftechnik Gmbh and a 100 kN load cell at room temperature. Two C-ring samples (Appendix A) were machined from each mesocarp with a thickness (b) ranging from 9 to 13 mm. Altogether, five mesocarps were used for machining specimens in the longitudinal direction, and five additional mesocarps were used for extracting specimens in the latitudinal direction. For each mesocarp, one specimen was tested in tension and one in compression test. The surfaces of the samples were ground with sandpaper (grits #80, #220, #500), and the sharp edges and corners were broken with grit #200 sandpaper to avoid stress concentrations. Epoxy resin was molded around the regions where the compression and tensile forces were applied to ensure improved force distribution. An amount of 10 C-ring samples (five longitudinal and five latitudinal) were then compressed between two parallel plates (Appendix A). Another 10 C-ring samples (five longitudinal and five latitudinal) were pulled by metal rings attached to the universal testing machine (Appendix A).

For all tests, a cross-head speed of 0.2 mm/min was used up to a pre-load of 5 N, and a cross-head speed of 0.5 mm/min was used after that. Elastic modulus and fracture strength were calculated from isotropic elastic beam theory for a semi-circular beam-shaped specimen with a rectangular cross-section, according to equations available in the Appendix A.

### 2.3. Compression Test

Mesocarp samples were cut in the longitudinal and latitudinal directions using a band saw (Appendix A). The cut surface was ground with sandpaper (grits #80 and #220). The specimen’s nomenclature was based on loading direction, following the same pattern as the C-ring specimen. Latitudinal loaded specimens (Appendix A) were cut from peduncle to opercular opening, resulting in two symmetric specimens. Differently, longitudinal loaded specimens were latitudinally cut, leading to specimens with an opercular opening (Appendix A) or a peduncle (Appendix A) at the top of the sample.

The compression tests were made using the same machine as for the C-ring tests. The test was monitored with shearography, which demands a low strain rate. Therefore a particular compression test procedure with different stages was applied: in the pre-load stage, until 10 N, the cross-head speed was 1 mm/min; in the initial slow stage (monitored with shearography), ranging from 10 N to 2 kN, a cross-head speed of 0.06 mm/min was used; this was followed by a fast stage with a cross-head speed of 5 mm/min, from up to 10 kN for the longitudinal specimens or up to 20 kN for the latitudinal specimens; and a final slow stage with a cross-head speed of 1 mm/min again monitored with shearography was used until failure. Appendix A shows a schematic force versus displacement curve explaining this procedure.

### 2.4. Shearography

Shearography or speckle shearing interferometry is a full-field speckle interferometric technique, which is surface displacement sensitive [24]. The optically rough surface of the object of interest is illuminated with laser light to form a speckle pattern (Appendix A). A shearing device combines the speckle pattern with an identical but laterally displaced version of itself. Images are recorded before and after a loading event, and the correlation of these images results in a fringe pattern that gives information on the displacement derivative on the surface [24]. When there is an internal defect in a loaded structure, the surface above this defect moves differently from its surroundings, which allows for the observation of internal defects and cracks in the shearography image. Appendix A shows that gray value gradients and discontinuities indicate bending stresses and cracks, respectively.

We used shearography to monitor the development of defects and cracks during mechanical testing of mesocarp semi-shells with a shear of $\vec{S}$ = 0°, which tracked the relative movement of rotation in and out of the plane in the Y-axis (Appendix A). The surface of each specimen was painted with white chalk to improve image quality.

The images were produced using Instra 4D software from Dantec Dynamics A/S (Skovlunde, Denmark) and processed with ImageJ 2.0 [25]. The use of the shearography method without interruption of the mechanical test is only possible when adopting very low strain rates. Each shearography image shows the relative rotation movement in the Y-axis (Appendix A) that occurred in 20 s during the mechanical test, representing a cross-head displacement of 0.02 mm in the compression test. With proper calibration and image processing, it is possible to obtain quantitative results of strain in the specimen; however, this goes beyond the scope of this work. Shearography was used to observe cracks and defects qualitatively.

### 2.5. Microstructural Analysis

The fractured specimens were observed by a VHX 100 light microscopy from Keyence and by an Inspect S 50 scanning electron microscope from FEI (Hillsboro, OR, USA) at 20.00 kV, SE mode, and working distances of 9–10.5 mm. Specimens were also photographed by a digital camera D3300 from Nikkon (Tokyo, Japan).

### 2.6. MicroCT

Non-tested mesocarp and fractured specimens of the compression tests were scanned with a Phoenix Nanotom m180 kV/20 W X-ray microCT and nanoCT Computed Tomography System from GE Measurement & Control. The radiographs were made with a tube voltage of 160 keV, a current of 110 µA, and a voxel size of 33 µm. The definition of a region of interest, segmentation, and further analysis was made with Fiji—ImageJ 2.0 [25] and the plugin BoneJ [26].

The void content of each specimen was estimated from several volumes of interest (VOI) evenly positioned inside of each of the mesocarp structures. Each VOI had a square cross-section with edge lengths of 0.66 cm or 0.51 cm depending on specimen thickness and contained a maximum of 250 slices. First, the images were normalized with a saturated value of 0.4% and filtered with a median 3D (radius 2) filter and a bandpass filter from 3 to 40 pixels. The images were then segmented with a user-defined threshold to distinguish the voids from the mesocarp cells. The void content was calculated with Equation (1) using the volume fraction function of the BoneJ plugin.
Void content (%) = (void volume)/(VOI) × 100(1)

The 2D radiographs, 2D segmented images, and 3D models of the volume of interest in mesocarp thickness are shown in Appendix A.

An ANOVA analysis was performed to evaluate whether the average values of void content were statistically different at a confidence level of 95% (α = 0.05).

### 2.7. FEM Analysis

The compression test of half mesocarp of *Bertholletia excelsa* was simulated with the finite element model (FEM) based on a mesh developed from microCT data. A mesocarp was scanned as described before with an approximate voxel size of 60 μm. The image stack was cropped, scaled (factor of 0.25), and edited with Fiji-Image J 2.0 to contain only the mesocarp shell with a resolution of 425 × 429 pixels. The user-defined threshold (between 110–255) segmented the images so that the white areas (comprising cells, seeds, and teguments) could be meshed and the black regions (comprising voids) be left empty. A filter erosion–dilation method was applied to decrease noise and delete solitary points in the images. Further details on image processing are available in [27].

The image stack was meshed using the MathWorks MATLAB plugin: iso2mesh (Natick, MA, USA). The mesh of tetrahedral elements was repaired and cleaned with Hypermesh. At this stage, the tegumented seeds were removed from the mesh so that only the mesocarp remained. A convergence study was conducted in Abaqus CAE to verify the capabilities of different models with increasing number of the 10-node tetra elements. At this point, a simple elastic behavior based on oak wood porperties, with a Young’s modulus of 12 GPa and Poisson’s ratio of 0.3, was assumed [28]. Models with number of elements varying from 227,487 to 942,076 were tested considering a maximum load of 20 kN applied at a node in the peduncle. The resultant contour plot of stress distribution served as an indicator of the mesh quality assessment. At this convergent study, the maximum stress observed in the models varied from 1400 MPa to 2400 MPa between the models while computational time varied from 1.5 to 45 min. A 460,664-element model was selected due to a favorable trade-off between precision and computational efficiency.

FEM was conducted in Abaqus CAE and consisted of simulating compression of three half mesocarp specimens derived from microCT data: latitudinal—peduncle (model A), latitudinal—opercular opening (model B), and longitudinal (model C). FEM nodes were set to simulate static, linear elastic behavior for a 10 kN force applied over the mesocarp surface, as shown in Appendix A.

For the material model, a Young’s modulus of 1070 MPa and Poisson’s ratio of 0.35 were assumed based on values given for mesocarp in previous mechanical testing [23]. The compression plate/mesocarp surface contact was defined as shown in Appendix A. It comprised surface triangles localized in a circular region around the compression plate center with a diameter of 40 mm and deviating 15° from the axial loading plane. The boundary conditions comprised a belt of nodes fixing the mesocarp equator (models A and B) or fixing the longitudinal section area (model C). The three models were evaluated for stress distribution and maximum stress/strain components.

## 3. Results

### 3.1. C-Ring Tests

The elastic moduli (E) and fracture strengths (σf) measured in the tensile and compressive C-ring tests are shown in Table 1. Longitudinal specimens exhibited higher elastic moduli (E~4 GPa) and fracture strengths (σf~26 MPa) than latitudinal ones. Interestingly, the stiffness under tensile and compressive loading is similar. These values are expected since longitudinal specimens have fibers with a favorable orientation in the most stressed regions during C-ring bending. These aspects are better explained in Appendix A.

Figure 2a,b highlight the crack path through mesocarp thickness in latitudinal and longitudinal specimens after tensile and compression tests, respectively. In the latitudinal specimens, fibers in layer I and layer III are orthogonal to the loading direction while fibers in layer II are parallel to it. Fiber bundles in layer II have to be broken during a fracture (intracellular cracks), promoting a tortuous crack path with intense deflection (red arrows in Figure 2a). In layers I and III, the main crack propagates through a fiber bundle interface or through the lighter regions involving bundles (pink arrows in Figure 2a) consisting of sclereid cells (intercellular cracks), which also results in a tortuous crack path (blue arrows in Figure 2a).

The opposite fiber orientation of longitudinal specimens leads to the different crack paths shown in Figure 2b. The breaking of fibers (intracellular crack) and crack deflection are seen in layers I and III (blue and green arrows in Figure 2b). In layer II of the longitudinal specimens, the debonding of fiber bundles from intercellular cracks passing through sclereids (pink arrows) or fiber bundles interface is observed (red arrows in Figure 2b). The structural voids of mesocarps are also indicated by arrows in Figure 2a,b. The voids near the main crack path can interact with the crack tip during a fracture.

It is noteworthy how the most stressed areas, especially the outer surface in layer III (Figure 2), are rough with many dimples and folds that can work as stress raisers assisting crack nucleation.

Figure 3 shows SEM images of the fracture surfaces and the crack paths of the C-ring specimens. The fracture surface, shown in Figure 3a, has regions where cracks propagated through the fiber’s interfaces, inside a bundle, or where fibers were debonded (blue arrows). In the same figure, red arrows highlight many broken fiber bundles, indicating the occurrence of fiber bridging and fiber pullout. Figure 3b shows one of these bundles in higher magnification. Fiber cells had a brittle failure, and some sclereid cells remained attached to the bundle, indicating a good adhesion between them.

Figure 3c shows an intercellular crack separating fibers inside a bundle. This crack path through fiber interfaces (compound middle lamellae) is shown in detail in Figure 3d. The middle lamellae show signs of plastic deformation and fibril bridging after fiber debonding, as highlighted by the green arrows in Figure 3d, indicating a good adhesion between fibers. Figure 3e shows an intercellular crack around a fiber bundle passing through the interface of sclereids and the bundle. It is noticeable that sclereid cells were not broken or torn during fracture. Figure 3f shows the main crack interacting with a structural void. This tubular structural void has remnants of vascular cells inside it and, apparently, was able to branch the main crack, contributing to energy dissipation.

### 3.2. Compression of Half Mesocarp

The compression tests of half mesocarps adopted a stage with fast strain rate to induce damage and stages with slow strain rates to allow for crack monitoring with shearography before and after damage. Typical force-displacement curves for longitudinal and latitudinal specimens are shown in Figure 4a. The longitudinal specimens supported higher forces and displacements than latitudinal specimens. The average values of maximum force and displacement at fracture at both directions are summarized in Table 2.

The fractured latitudinal and longitudinal specimens are shown in Figure 4b,c, respectively, where the main crack is highlighted. Although many cracks and defects were monitored with shearography during the test, the final failure was caused by one main crack, which started near the upper compression plate and propagated towards the base plate. The shearography monitoring of latitudinal (Figure 4d) and longitudinal (Figure 4e) typical specimens exemplify the crack development in both directions.

During the first slow loading stage (step (I)—Figure 4d,e), shearography indicates local deformation concentrations. These are most probably voids because cracks due to testing are improbable at these low loads (from 10 N to 2 kN). Although we can assume that the structural voids are similar in both specimens, they are more evident in Figure 4d. In step (II), the latitudinal specimen has a large crack that quickly grew during the fast stage while the longitudinal specimen has just a few small cracks, which slowly evolved during the subsequent slow stage. Additionally, longitudinal specimens had higher deformation than latitudinal specimens, as indicated by the dashed white and red lines in step (III) of Figure 4d,e.

Interestingly, most voids are not visible after the fast-loading stage, as if all the relative movement or deformation was concentrated near the main cracks formed during the fast-loading stage (arrows in steps II and III in Figure 4d,e).

Figure 5 shows pictures of the interior and the outside of the fractured specimens, where the main crack is pointed up. The opercular opening and peduncle are the main features of the mesocarp interior. The structural ridges, indicated by red arrows in the longitudinal–peduncle specimen in Figure 5a, consist of a great number of large fibers aligned with the mesocarp longitudinal section.

The main crack is parallel to the load direction in all specimens. As mentioned before, longitudinal specimens were severally deformed, leading to specimens with a flat top (blue circle in Figure 5a,b). The shell thickness was bent during compression, and the most stressed region, where the main crack probably nucleated (see Figure 4e), is delimited by the blue circle in Figure 5a,b. This flat top was not observed in the latitudinal specimen in Figure 5c.

Figure 6 shows the main crack path through mesocarp layers. In longitudinal specimens (Figure 6a), the main crack path is parallel to fiber bundles in layers I and III while it is orthogonal to these cells in layer II. The opposite occurs in latitudinal specimens (Figure 6b), where the main crack path is parallel to fibers in layer II and orthogonal to fibers in layers I and III. The main crack breaks the orthogonal fiber bundles while propagating through sclereids (pink arrows) or fiber bundles interface when fibers are parallel to its path.

The void content before and after the compression test, estimated with microCT, is shown in Table 3. ANOVA analysis showed no significant difference in the void content between the non-tested specimen and fractured specimens. Therefore, previous voids did not change during the compression test. Appendix A shows similar 3D models of voids from the volume of interest (VOI) defined in the non-tested, latitudinal, and longitudinal specimens.

The microCT analysis of the fractured specimens allowed for the investigation of cracks other than the main one. The longitudinal specimen (3D model shown in Figure 7a) had small cracks starting at the inner surface of the opercular opening, as shown in Figure 7b. They also have a radial propagation path through the shell thickness. These cracks are smaller than the main crack and did not reach the outer surface. Apart from them and the main crack, this longitudinal specimen had two further expressive cracks, both with similar behavior. One of them is shown in Figure 7c,d. This crack nucleated at the outer surface, where the shell experienced intense bending and propagated mostly toward the specimen base. At the point indicated by the red line in Figure 7a, this crack had not broken the entire mesocarp thickness, as shown in Figure 7c. Figure 7d shows the region where all mesocarp layers were broken at the point indicated by the green line in Figure 7a. This crack path ended near the yellow line in Figure 7a.

Mesocarps have a dense net of structural voids seen as black marks in the microCT 2D slices. Although there is a resemblance between cracks and voids, cracks are sharper, thinner, and considerably more tortuous than structural voids. Interestingly, there are numerous structural voids in layer I (see green arrows in Figure 7c), which can interact with the crack tip. In Figure 7d, the crack seems to be attracted to different voids, connecting them to what leads to a ramified and tortuous crack path.

Figure 8 shows a microCT 3D model and 2D slices of a latitudinal specimen where the main crack was the only one to pass through the entire mesocarp thickness. The positions of each 2D slice along the main crack path are indicated in Figure 8a. Figure 8b shows the main crack (highlighted in red) near the inner surface of the mesocarp. The main crack ramified when it reached the volume below the loaded surface. At this position, no cracks are seen on the outer surface. Figure 8c shows that all mesocarp layers were broken by the main crack in a place near the top of the specimen (red line in Figure 8a). Thenceforth, the main crack path is towards the specimen base.

Structural voids can nucleate or interact with secondary cracks, as shown in Figure 8d, where the crack (highlighted in red) is perpendicular to a tubular void (green arrow). However, such interaction between secondary cracks and structural voids was not common (see voids net in Figure 7 and Figure 8).

### 3.3. FEM Analysis

The compression test was simulated with a finite element model (FEM) from microCT data of three half mesocarp specimens: longitudinal–peduncle (model A), longitudinal–opercular opening (model B), and latitudinal (model C). The von Mises stress contour plots for all three models are shown in Figure 9.

Model A exhibited low overall Mises stresses on the outer surface and relatively uniform distribution. Mises stress was mainly concentrated in the structural ridges, where a maximum value of 56 MPa was observed. Contour plots along the mesocarp inner surface between the ridges displayed values of Mises stress under 25 MPa, showing that the internal ridges absorbed most of the load.

Model B has a raised opercular opening that supported most of the forces while stress quickly propagated inside the mesocarp. The maximum Mises stress of 92 MPa was read at the inner edge of the opercular opening inside the loaded surface. Apart from the loaded surface, most of the mesocarp outer surface shows low stress values. Like Model A, the Mises stress distribution was more intense along the structural ridges inside the shell.

Model C displayed a different Mises stress distribution. Mises stress levels over 30 MPa and a maximum of 95 MPa were read in a large and spherical volume below the loaded surface. The structural features were not so requested in this test configuration as in models A and B. Only the inner ridges directly below the loaded surface absorbed the load while ridges near it showed little to no stress. Mises stress levels above 20 MPa were read in a large area inside the mesocarp while regions near the boundary fixation, i.e., the peduncle and opercular opening, displayed Mises stress around 10 MPa.

Regarding principal stresses, Model A showed a maximum tensile stress of 59 MPa at the structural ridges. The loaded surface around the peduncle with diameters of 40 mm and approximately 30 mm, respectively, experienced compressive stresses of up to 14 MPa. This compression load bent the mesocarp inward, exerting tensile stress on the inner regions of the mesocarp, especially on the ridges. Model B displayed compressive stresses as high as 40 MPa in the wall and the crease around the opercular opening while folds and dimples on the outer surface had tensile stresses of up to 46 MPa. The volume below the loaded surface in Model C displayed compressive stresses (87 MPa) on the outside and tensile forces (112 MPa) on the inside.

## 4. Discussion

When the specimens are bent in the C-ring tests, the maximum tensile stresses occur on the internal surface (tensile loading) or on the outer surface (compression loading) [29]. Therefore, this test evaluates the strength based on the distribution of flaws on the surfaces of layer I and III, respectively [30]. *Bertholletia excelsa* mesocarps have a rough surface with lots of folds and dimples (Figure 5) that can work as stress raisers during the test and facilitate crack nucleation.

A tortuous crack path was observed in all mesocarp layers regardless of its preferential fiber orientation (Figure 2). Fiber pullout, fiber bridging, and crack deflection are expected when fibers are orthogonal to the crack path [3,31,32]. However, the tortuous crack path through parallel fibers (see layer II of Figure 2b) draws attention. Figure 10a,b shows a schematic drawing of the crack path through different fiber orientations. The mesocarp’s unique microstructure, where fibers are grouped as bundles involved by sclereids cells (Figure 1c), promotes such a crack development. In previous work, the micromechanical tests of mesocarps showed intercellular cracks propagating on the interface between sclereids and around fiber bundles [23]. The same crack development was observed in the layer II of Figure 2b in the C-ring tests. A path through sclereid interfaces requires less energy than separating or breaking a densely packed bundle of fibers but results in a tortuous crack path, which is a desirable feature for high fracture toughness materials.

SEM images (Figure 3) provided evidence for other toughening mechanisms. Some fiber bundles experienced fiber pullout and fiber bridging before brittle failure from an intracellular crack, as shown in Figure 3a,b. The intercellular cracks through interfaces, i.e., middle lamellae, showed fibril bridging and plastic deformation (Figure 3d). These energy dissipation mechanisms of the middle lamellae were also observed in the micromechanical tests of the mesocarp [23]. Moreover, crack branching was observed when cracks interacted with structural voids (Figure 3f).

The stress distribution and crack development in the compression test of the half mesocarp are more complex. The FEM analysis indicated that the volume below the loaded surface under compression loads is bent inward, exerting tensile stress on the inner regions. Such a stress distribution explains the radial cracks in the inner surface, which did not reach the outer surface, seen in microCT 2D slices of both kinds of specimens (Figure 7b and Figure 8b). In latitudinal specimens, this radial crack in the inner surface propagated towards the specimen base, resulting in the main crack path parallel to the load. The formation of a flat top in longitudinal specimens (Figure 5a,b) implies shell bending, which induces tensile stress on the outer surface and compressive stress on the interior surface. This stress state can cause crack propagation from layer III to layer II, shown in Figure 7c. Eventually, these cracks reached layer I, crossing all mesocarp thickness (Figure 7d) while propagating towards the sample base. Only one of these cracks developed into the main crack by reaching the specimen base. Therefore, the main crack in both specimen orientations nucleated in bent regions and propagated parallel to the load, from top to base (Figure 4d,e and Figure 5). As a result of this crack path, layer II played a more significant role in the mechanical performance than occurred in the C-ring tests. This is very significant since it is the thickest layer of mesocarps and contains the highest and the most densely packed number of oriented fibers [20].

The fiber bundle orientation, the structural ridges, and the specimen geometry together determine the differences seen in maximum loads and deformations for the different orientations. According to FEM, the structural ridges were the main features sustaining and distributing the load under the elastic regime, functioning as a skeleton for the mesocarp. It also showed that the rough outer surface, with its folds and dimples, are stress raisers, as was also indicated by the C-ring results.

In the compression of the half mesocarp, longitudinal specimens sustained higher loads and showed higher deformation and slower crack development (Figure 4a,e). As illustrated in the FEM of these specimens (Figure 9—models A and B), the structural ridges absorbed most of the stress and distributed it more uniformly. In latitudinal specimens (Figure 9—model C), the stress is concentrated in the region below the loading plate. In this specimen configuration, just a few ridges right below the loaded area are stressed, which compromises a uniform stress distribution. These results show that, during the elastic regime, the geometry of the fruit and its structural inner ridges better distribute compression loads applied perpendicular to its latitudinal section. Additionally, the fiber orientation throughout the mesocarp relative to the loading diretion explains the higher deformation with the formation of a flat top in the longitudinal specimen. Figure 10 shows a schematic drawing of both specimen configurations. The red lines represent fibers in layer II while blue and green lines illustrate the fibers in layer I (including the structural ridges) and III, respectively. The volume below the upper loaded surface, which was severely deformed, is highlighted in pink. In longitudinal specimens (Figure 10c), only fibers in layers I and III can hamper shell bending and flat top formation. The fiber orientation in layer II, which consists of the most mesocarp volume, is not helpful, leading to more intense deformation. Both fiber orientations in latitudinal specimens (Figure 10d) can restrain shell bending and flat top formation. However, only bundles right below the loaded surface were affected due to the ineffective stress distribution in this specimen configuration.

The crack development of the differently oriented specimens is also explained by the fiber orientation throughout the mesocarp relative to the main crack path (Figure 10). Shearography images (Figure 4d,e) and microCT slices (Figure 7a and Figure 8a) indicated that most cracks nucleated in the most stressed region (pink areas in Figure 10) due to shell bending. Assuming that the main crack developed through the shell bulk towards its base, the densely packed fibers of layer II in longitudinal specimens (in Figure 10c) can better constrain the crack propagation. Figure 6a shows numerous fibers in the thick layer II, which were broken during the fracture at a great energetic cost. This explains the slower crack development observed during the compression test (Figure 4e).

The shearography analysis of latitudinal specimens revealed faster crack development, resulting in failure at lower maximum loads and displacements (Figure 4a,d). In this configuration (Figure 10d), layer II is a weaker obstacle to crack propagation as the crack can easily run through the fiber interfaces, a more straightforward and weaker path as shown previously [23]. Nonetheless, a tortuous crack path through this fiber orientation was also observed (Figure 6b). Like C-ring specimens, intercellular cracks propagate around bundles, mostly through sclereids, which is an interesting energy-consuming mechanism. Figure 10a,b illustrate such a tortuous crack path. Only fiber bundles in layers I and III in latitudinal specimens were advantageously oriented to hamper crack propagation by dissipating energy during fiber breakage. However, Figure 6b shows that these layers are considerably thinner than layer II with more dispersed fiber bundles involved by sclereid cells.

The better mechanical performance of longitudinal loaded specimens compared to latitudinal loaded specimens was also observed in compression tests of whole mesocarps of *Bertholletia excelsa* [17]. This suggests that the mechanisms described above also occur in the fracture of whole fruits.

Another interesting aspect is that the void content did not increase significantly during compression. The microCT 3D models of voids of non-tested and compressed specimens are similar, and the microCT 2D slices from fractured specimens show that only a few structural voids, in fact, interact with the cracks.

We assume that only the most severe defects, probably in the most critically mechanically affected volume, can developed into growing cracks during the test. Moreover, voids are not visible with shearography after the fast stage, as if the stress fields were not uniform anymore but concentrated near the main cracks. After crack nucleation, structural voids can attract cracks, leading to deflection, ramification, and tortuous crack paths (Figure 3f, Figure 7c,d and Figure 8). Previously, the monitoring of micromechanical tests evidenced a significant interaction between structural voids and the crack tip, resulting in the connection of pre-existing voids and crack deflection [20].

## 5. Conclusions

In this study, we investigated how fruit geometry and fiber orientation affect the macro-mechanical properties of *Bertholletia excelsa* mesocarps. C-ring tests were used to determine the elastic modulus and the fracture strength in tension and compression, and compression tests of half mesocarps yielded the maximum (fracture) force and maximum displacement at the fracture. Crack development during the compression test was monitored with shearography while an FEM analysis allowed for an evaluation of the stress distribution. Measurements of the void contents before and after the compression of the hemispheres by microCT helped us to evaluate microstructural damage in the bulk in greater detail. Additionally, SEM and digital microscopy allowed for the observation of cracks through different cell morphologies and preferential orientations. The main remarks are stated below:
The elastic modulus of C-ring specimens perpendicular to the mesocarp latitudinal section (~4 GPa) is twice the modulus measured parallel to it. This anisotropy is caused by the preferential fiber orientation of layers I and III, the most stressed regions during bending.The fracture strength can reach 26.4 ± 3.6 MPa in the compression of C-ring longitudinal specimens. The rough mesocarp surface with folds, dimples, and many stress raisers can easily nucleate cracks during C-ring tests and when decreasing specimens’ mean strength.Fiber bundles orthogonal to the crack path show an intense crack deflection and experienced fiber pullout and fiber bridging before brittle failure, dissipating a great amount of energy. Surprisingly, fiber bundles oriented parallel to the crack path also contibuted to the energy dissipation. Intercellular cracks propagating through fibers and sclereids’ middle lamellae promoted a long and tortuous crack path. After the debonding of fiber bundles, plastic deformation and fibril bridging was observed in the middle lamellae.The fiber bundle orientation, the structural ridges, and the specimen geometry together explain the high forces and displacements of longitudinal specimens in the compression test of half mesocarps.The FEM analysis indicated that the structural ridges uniformly distributed the stress in the shell only when oriented parallel to the load direction (longitudinal specimens). The fiber and ridge orientations of longitudinal specimens also promoted intense deformation with the formation of a flat top, resulting in the bending of the shell and crack nucleation.Crack development in longitudinal specimens was slowed by the breaking of numerous fiber bundles in layer II, positioned orthogonal to the crack path.Latitudinal specimens showed stress concentrations in a few ridges below the loaded surface, where the main crack nucleated. Fiber orientation in mesocarp layer II is parallel to the crack path in latitudinal specimens, resulting in the faster crack propagation observed in these specimens.The void content did not considerably change after the compression tests, indicating that only voids in the most critical mechanically affected volume developed into growing cracks. However, these structural voids interacted with cracks, mainly attracting, deflecting, and ramifying them.


In conclusion, the understanding of the *Bertholletia excelsa* fruit strategies to overcome a challenging opening can provide insights for fiber-reinforced composites or new designs for protective capsules, vessels, or devices.

## Figures and Tables

**Figure 1 biomimetics-08-00509-f001:**
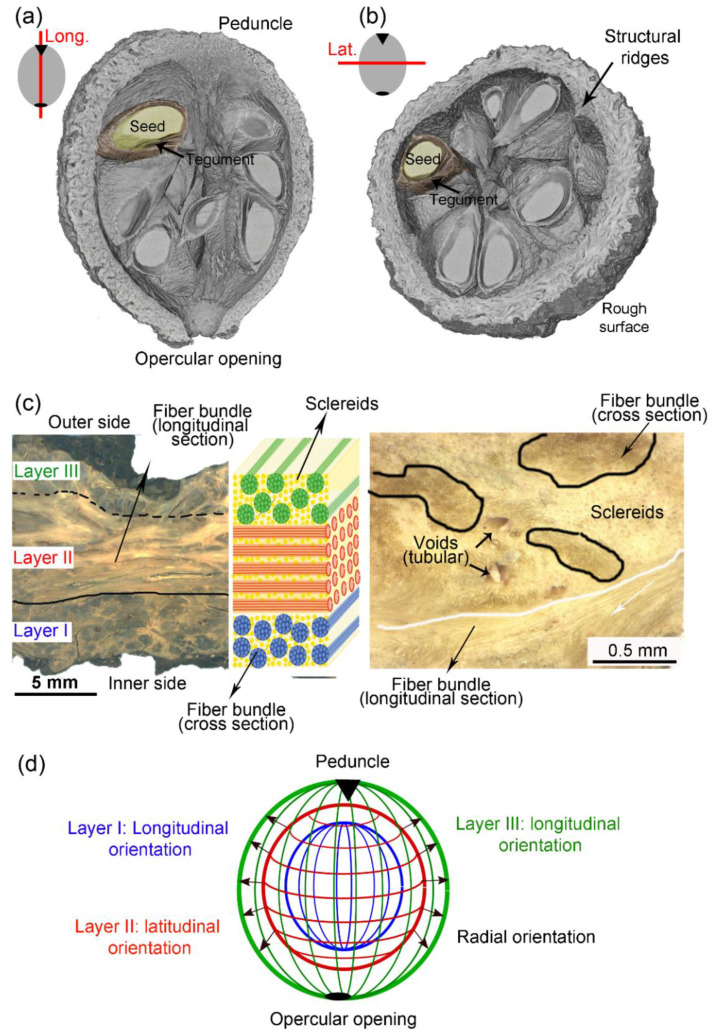
*Bertholletia excelsa* mesocarp: microCT volume reconstruction showing (**a**) longitudinal and (**b**) latitudinal sections; (**c**) light micrographs of a latitudinal section and schematic drawing showing the three mesocarp layers; (**d**) schematic drawing of the preferential orientations of fiber bundles in the three layers identified in the mesocarp.

**Figure 2 biomimetics-08-00509-f002:**
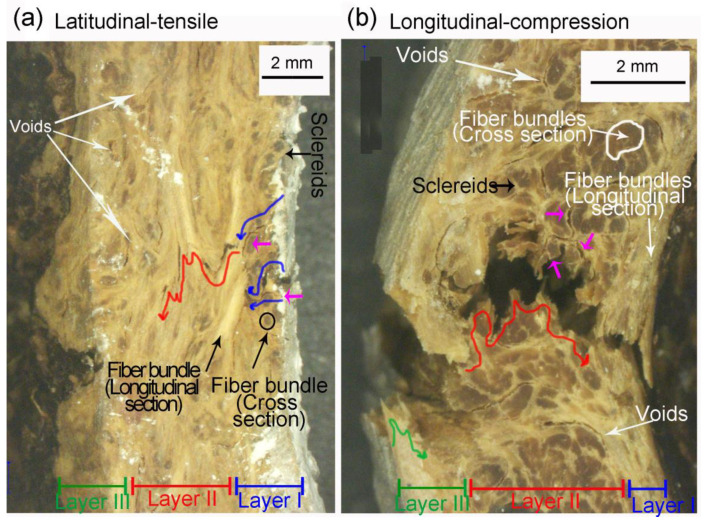
C-ring fractured specimens: (**a**) latitudinal specimen after tensile loading and (**b**) longitudinal specimen after compression loading. Blue, red, and green arrows indicate the main crack path in layers I, II, and III, respectively. Pink arrows indicate crack path through sclereid cells.

**Figure 3 biomimetics-08-00509-f003:**
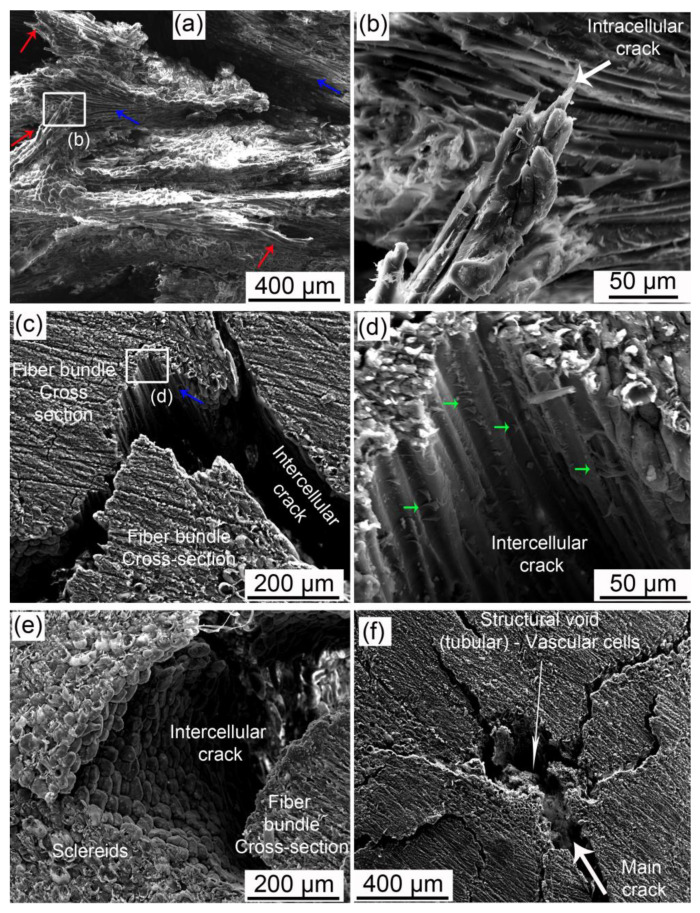
SEM images of C-ring fractured specimens. Fractured surface: (**a**) broken (red arrow) and debonded (blue arrow) fiber bundles; (**b**) fractured fiber bundle (intracellular crack) is shown in higher magnification. Lateral view of crack path: intercellular cracks through (**c**) fibers interface inside a bundle, (**d**) middle lamellae, and (**e**) between sclereids and a fiber bundle; (**f**) crack interaction with a structural void. Red arrows indicate where fiber bundles were pulled-out and broken, blue arrows highlight intercellular cracks inside fiber bundles or debonding, and green arrows point to signs of plastic deformation of the middle lamellae during debonding.

**Figure 4 biomimetics-08-00509-f004:**
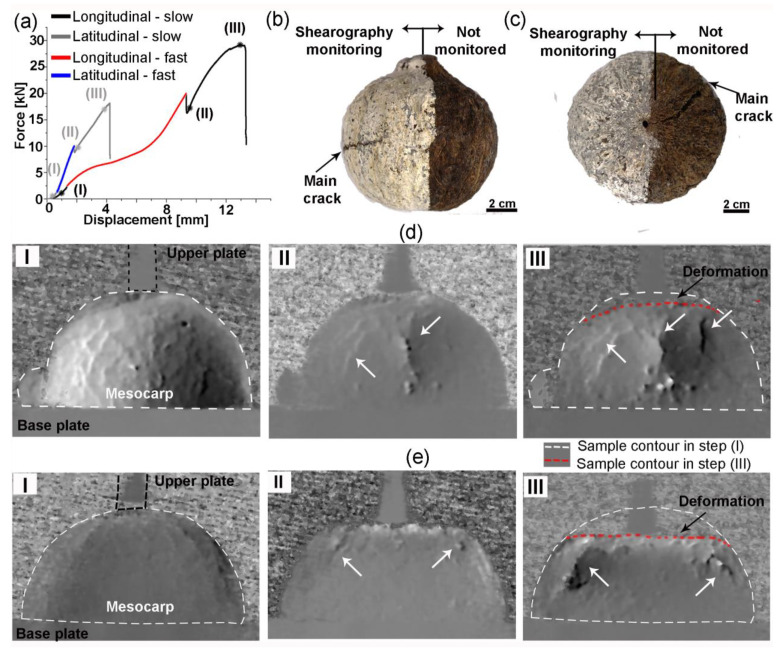
Compression of half mesocarp: (**a**) force versus displacement curve where steps I, II, and III are highlighted by asterisks; (**b**) mesocarp surfaces of latitudinal and (**c**) longitudinal specimens after compression testing; shearography monitoring of (**d**) latitudinal and (**e**) longitudinal specimens. In steps I and III, arrows highlight crack development during the test, and dashed white and red lines delimit the sample shape. A black dashed line delimits the upper compression plate.

**Figure 5 biomimetics-08-00509-f005:**
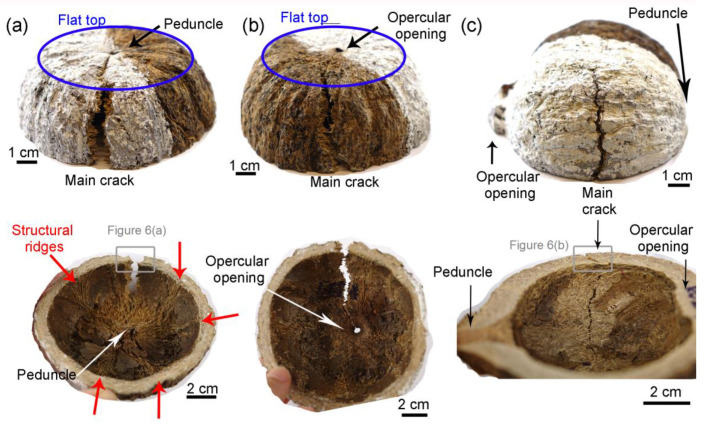
Pictures of the interior and the outer surface of fractured half mesocarps: (**a**) longitudinal with peduncle, (**b**) longitudinal with opercular opening, and (**c**) latitudinal specimens. Mean features, including structural ridges (red arrows) and main crack, are highlighted. The blue circle indicates the flat specimen top and the most stressed areas due to the bending of the shell during compression.

**Figure 6 biomimetics-08-00509-f006:**
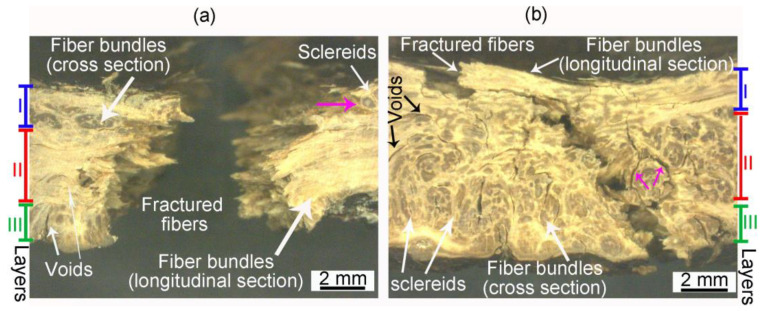
Micrographs of the main crack in the base of the (**a**) longitudinal and (**b**) latitudinal specimens. Fiber bundles (longitudinal and cross section), sclereids, voids and layers I (blue), II (red), and III (green) are highlighted. Pink arrows indicate a crack path through sclereids.

**Figure 7 biomimetics-08-00509-f007:**
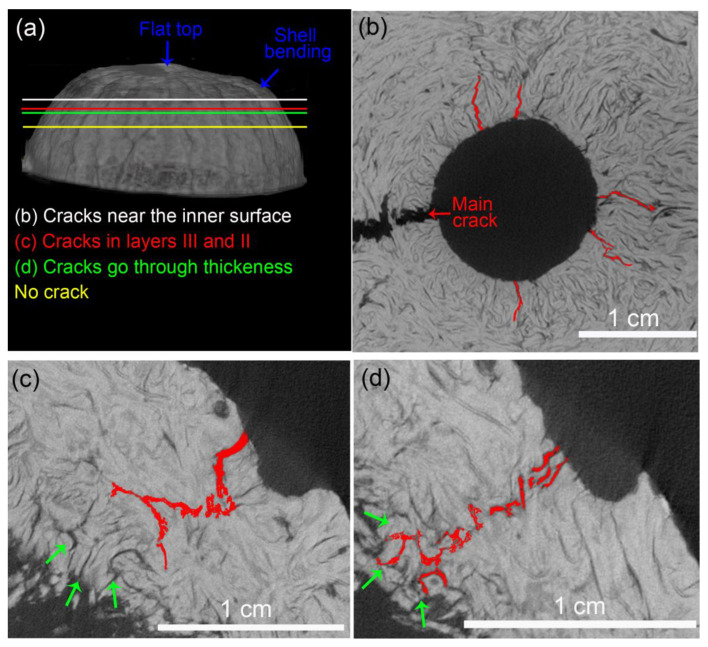
MicroCT analysis of secondary crack paths (highlighted in red) in longitudinal specimen: (**a**) 3D model showing the positions of the 2D slices, which shows (**b**) cracks around the opercular opening near the inner surface of mesocarp (white line in (**a**)) and one crack at different positions (**c**) propagating through layer III and II (at the red line in (**a**)) and (**d**) through the entire thickness (green line in (**a**)). This crack stopped before the yellow line in (**a**). Green arrows indicate structural voids.

**Figure 8 biomimetics-08-00509-f008:**
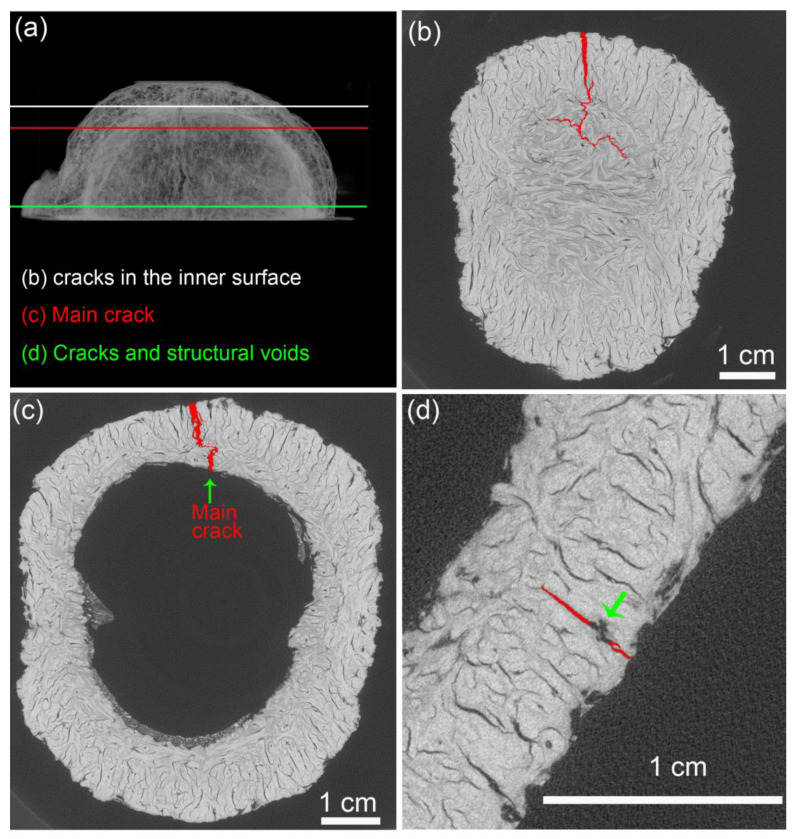
MicroCT analysis of crack paths (highlighted in red) in latitudinal specimen: (**a**) 3D model showing the positions of the 2D slices which shows the main crack (**b**) in the inner surface below the loaded area and (**c**) passing through mesocarp thickness, and a (**d**) secondary crack nucleated in a tubular structural void (green arrow). Crack paths are highlighted in red.

**Figure 9 biomimetics-08-00509-f009:**
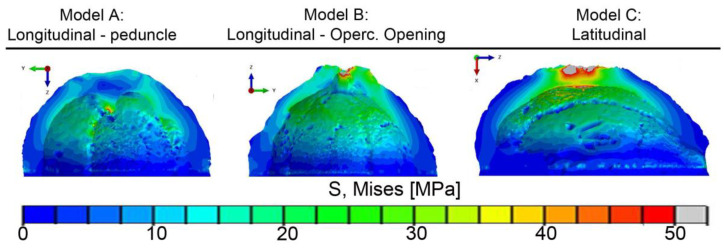
Von Mises stress contour plots showing the stress distribution from FEM analysis of compression tests on longitudinal specimen containing peduncle (model A), opercular opening (model B), and on a latitudinal specimen (model C).

**Figure 10 biomimetics-08-00509-f010:**
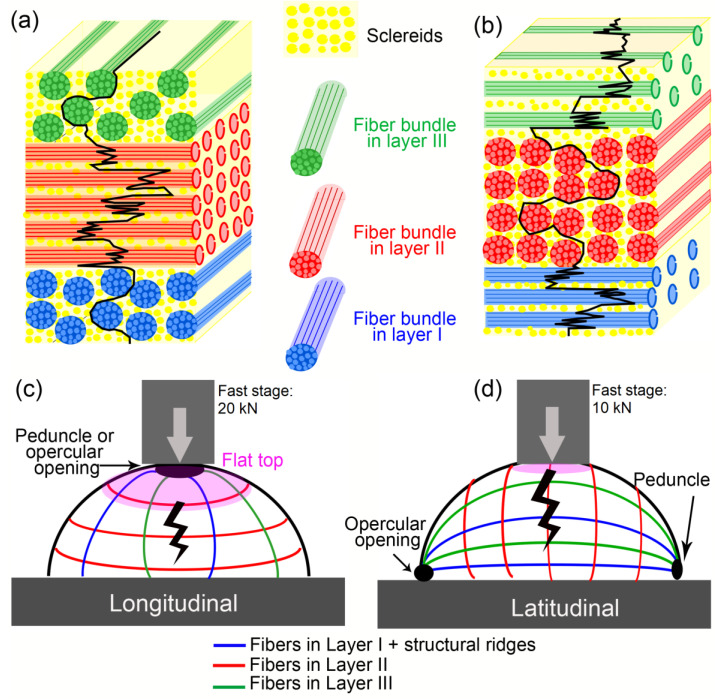
Schematic drawing of crack path at different fiber orientations in C-ring (**a**) latitudinal and (**b**) longitudinal specimens and the fracture mechanism during compression test of half mesocarp considering the orientation of fibers for (**c**) latitudinal and (**d**) longitudinal specimens.

**Table 1 biomimetics-08-00509-t001:** Elastic moduli and fracture strengths determined in the C-ring tests.

Force	Specimen	E [GPa]	σ_f_ [MPa]
Compression	Latitudinal	2.3 ± 0.3	20.1 ± 3.8
	Longitudinal	4.0 ± 1.0	26.1 ± 3.9
Tension	Latitudinal	2.0 ± 0.3	16.2 ± 3.3
	Longitudinal	3.7 ± 0.5	26.4 ± 3.6

**Table 2 biomimetics-08-00509-t002:** Maximum force and displacement at fracture for latitudinal and longitudinal half-spheres under compression.

	Longitudinal	Latitudinal
Maximum force [kN]	26.0 ± 2.8	15.6 ± 3.3
Displacement at break [mm]	9.8 ± 3.3	4.5 ± 1.0

**Table 3 biomimetics-08-00509-t003:** Void content of non-tested mesocarp and of fractured half-spheres loaded in compression.

Specimen	Void Content [%]
Non-tested mesocarp	12.8 ± 2.1
Latitudinal specimen	13.7 ± 2.7
Longitudinal specimen–peduncle	12.7 ± 2.4
Longitudinal specimen–opercular opening	11.5 ± 2.8

## Data Availability

The datasets generated during and/or analyzed during the current study are available from the corresponding author on reasonable request.

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
