# Peer review of "Betholletia excelsa Fruit: Unveiling Toughening Mechanisms and Biomimetic Potential for Advanced Materials"

_biomimetics, 2023, doi:10.3390/biomimetics8070509_

Round 1

Reviewer 1 Report

Comments and Suggestions for Authors

The authors studied the influence of fruit geometry and fiber orientation onto the macro-mechanical properties of Bertholletia excelsa mesocarp. They used C-ring tests determine the elastic modulus and the fracture strength in tension and compression, and compression tests of half mesocarp to obtain the maximum (fracture) force and maximum displacement at fracture. FME was applied to determine the stress distribution. 

The paper is well structured. It would be of interest to the research community in this field of work. Some minor improvements should still be addressed: 

1) According to my information, there is no limit regarding the article length in MDPI journals. Hence, why not providing the figures from the supplementary material directly in the article? In this way, everything would be in one place, without the need to download different files. I would recommend adding all the important figures to the manuscript. 

2) In many places, references are given after the sentence, i.e. after the dot. This should not be done so. The reference, to which the sentence refers must be given in that sentence. 

3) The authors use the term "volume of interest" (VOI). The term should be explained. 

4) The mesh of tetrahedral elements had 100 px of volume, 2.5 px as a maximum size of the triangle's surface, and a threshold of 0.5. I assume px stands for pixel? Whatever it is actually, I find it strange that it is used to measure both volume and surface. This should be avoided. Regarding the FE mesh, please provide the information about the mesh in terms of number of degrees of freedom (DOFs), or number of elements, or number of nodes. 

5) Was the convergence analysis done with the FE mesh? Some of the results show properties characteristic for singularities in FE analysis. 

6) "FEM was analyzed in Abaqus CAE..." The part of the sentence is not well formulated. It should rather read: "The FE analysis has been done/conducted in Abaqus". 

7) "FEM nodes were set to simulate static, nonlinear, and elastic behavior..." The formulation needs to be better. An FE analyst would set that the FE mesh was set to... Another aspect is the mentioned nonlinearity. What kind of nonlinearity the authors refer to? Geometric or material? Whatever the selection was, hat was the reason for it? 

8) Make sure that subtitles start with a capital letter (check 3.2, for instance)

9) The scale for the von Mises stresses in Fig. 9 runs till the value of 50 MPa, but values exceeding 90 MPa are reported. Why did the authors select such a scale? 

10) Formatting of references should be done in a consistent manner. For instance, the authors use all capital letters in ref#23, which should definitely not be the case. This is not the only inconsistency with the the journal template. Also, there is only 1 reference from the previous 2 calendar years, which is not enough to provide the state of the art in the field. 

Author Response

Reviewer 1
The authors studied the influence of fruit geometry and fiber orientation onto the macro-mechanical properties of Bertholletia excelsa mesocarp. They used C-ring tests determine the elastic modulus and the fracture strength in tension and compression, and compression tests of half mesocarp to obtain the maximum (fracture) force and maximum displacement at fracture. FME was applied to determine the stress distribution. 

The paper is well structured. It would be of interest to the research community in this field of work. Some minor improvements should still be addressed: 

  • According to my information, there is no limit regarding the article length in MDPI journals. Hence, why not providing the figures from the supplementary material directly in the article? In this way, everything would be in one place, without the need to download different files. I would recommend adding all the important figures to the manuscript. 

While MDPI does not impose specific limits on the number of figures or manuscript length, both we and the other reviewers found the manuscript to be excessively lengthy. Consequently, we have made an effort to retain the most crucial figures within the main body of the manuscript and provide the less pertinent ones as supplementary information.

2) In many places, references are given after the sentence, i.e. after the dot. This should not be done so. The reference, to which the sentence refers must be given in that sentence. 

The manuscript was modified according to suggestion.

3) The authors use the term "volume of interest" (VOI). The term should be explained. 

The ImajeJ software handles the majority of quantitative image analysis within a user-defined region. In the case of microCT scans, users select an area within the image and specify the number of slices, thereby creating a region known as the Volume of Interest (VOI). In this work, areas/volumes with identical dimensions were selected from different positions across all analyzed samples. In the text, it is now explained by:

‘The void content of each specimen was estimated from several volumes of interest (VOI) evenly positioned inside of each of the mesocarp structures. Each VOI had a square cross-section with edge lengths of 0.66 cm or 0.51 cm, depending on specimen thickness, and contained a maximum of 250 slices.’

4) The mesh of tetrahedral elements had 100 px of volume, 2.5 px as a maximum size of the triangle's surface, and a threshold of 0.5. I assume px stands for pixel? Whatever it is actually, I find it strange that it is used to measure both volume and surface. This should be avoided. Regarding the FE mesh, please provide the information about the mesh in terms of number of degrees of freedom (DOFs), or number of elements, or number of nodes. 

We removed the description of elements using px, as suggested. We also included in the manuscript the information requested:

-Number of elements tetra mesh with 460,664 elements

-Number of nodes: 10-node tetra element

5) Was the convergence analysis done with the FE mesh? Some of the results show properties characteristic for singularities in FE analysis. 

We included a description of the convergence study in the manuscript. Resuming, we tested models with number of elements varing from 227,487 to 942,076, which resulting in computational time increasing from 1.5 to 45 minutes. The 460,664-element model was selected due to a favorable trade-off between precision and computational efficiency.

We did not observe singularities in the FE analysis Furthermore, the maximum stress levels recorded during the analysis were acceptable, primarily localized within regions anticipated for stress concentration, owing to the irregular geometry of the mesocarp.

6) "FEM was analyzed in Abaqus CAE..." The part of the sentence is not well formulated. It should rather read: "The FE analysis has been done/conducted in Abaqus". 

The sentence was altered as suggested.

7) "FEM nodes were set to simulate static, nonlinear, and elastic behavior..." The formulation needs to be better. An FE analyst would set that the FE mesh was set to... Another aspect is the mentioned nonlinearity. What kind of nonlinearity the authors refer to? Geometric or material? Whatever the selection was, what was the reason for it? 

The sentence was wrong, so the manuscript was modified.

We meant to say that the FEM was set to simulate linear elastic behavior. Although the nonlinear behavior is more adequate for the mesocarp material and geometry, its complexity would exceed the scope of this work.

8) Make sure that subtitles start with a capital letter (check 3.2, for instance)

All subtitles were checked.

9) The scale for the von Mises stresses in Fig. 9 runs till the value of 50 MPa, but values exceeding 90 MPa are reported. Why did the authors select such a scale? 

Based on our personal judgment, the scale was chosen to provide a clearer view of stress distributions in the mesocarp features.

10) Formatting of references should be done in a consistent manner. For instance, the authors use all capital letters in ref#23, which should definitely not be the case. This is not the only inconsistency with the the journal template. Also, there is only 1 reference from the previous 2 calendar years, which is not enough to provide the state of the art in the field

As per the suggestions, Reference #23 was corrected, and three references from 2023 were added to the document.

Reviewer 2 Report

Comments and Suggestions for Authors

Manuscript title: Betholletia excelsa fruit: unveiling toughening mechanisms and biomimetic potential for advanced materials

In the manuscript written by Sonego et al, the authors describe the mechanical testing followed by various analysis techniques (microCT, FEM) of the shells of brasil nut.

The introduction is quite extensive with a lot of references to the SI. I’d suggest to reduce it to the necessary parts and include a sketch of the fruit, so it’s easier for the reader to refer to.

For the results: in c-ring compression and tension, the distribution of crack/compression regions is well known. Try to reduce this part to a minimum and focus on the important results. The authors obtained many results, try to condense the part down to the most interesting one, think about moving some to the SI as the manuscript is to extensive and hard to follow. The number of figures should be reduced to a reasonable amount.

In the section talking about the compression of half mesocarp, the authors report maximum force in kN. For better comparability, report compression strength in MPa.

Otherwise, the discussion and conclusion are well written.

Comments on the Quality of English Language

Some minor errors, nothing problematic.

Author Response

Manuscript title: Betholletia excelsa fruit: unveiling toughening mechanisms and biomimetic potential for advanced materials

 In the manuscript written by Sonego et al, the authors describe the mechanical testing followed by various analysis techniques (microCT, FEM) of the shells of brasil nut.

The introduction is quite extensive with a lot of references to the SI. I’d suggest to reduce it to the necessary parts and include a sketch of the fruit, so it’s easier for the reader to refer to.

We made a concerted effort to streamline the introduction, retaining essential information for result comprehension. Additionally, in response to your suggestion, we've included a mesocarp figure.

For the results: in c-ring compression and tension, the distribution of crack/compression regions is well known. Try to reduce this part to a minimum and focus on the important results. The authors obtained many results, try to condense the part down to the most interesting one, think about moving some to the SI as the manuscript is to extensive and hard to follow. The number of figures should be reduced to a reasonable amount.

Indeed, we have many results and numerous figures. As suggested, we moved the discussion and the figure of C-ring crack/stressed regions to supplementary information. We also moved the image of  microCT 3D models of voids to supplementary information (Figure S7). However, we keep most of the figures in the manuscript since we believe that they are useful to show how mesocarp microstructure interacts with cracks, which is the main objective of the work.

In the section talking about the compression of half mesocarp, the authors report maximum force in kN. For better comparability, report compression strength in MPa.

Unfortunately, as mesocarp is a thick spherical structure we are unable to provide compression values in MPa, as there is no valid model or equation available for this.

Otherwise, the discussion and conclusion are well written.

Thank you!

Reviewer 3 Report

Comments and Suggestions for Authors

The paper “Betholletia excelsa fruit: unveiling toughening mechanisms and biomimetic potential for advanced materials” is interesting and centered on the scope of the journal. The paper is suitable to publish but some modifications are required.

Introduction

The introduction is well written, but I suggest to add a paragraph about to further explore the possible applications. It is not enough just to say that the work may be useful for composites filled with natural fibers.

Materials and methods

The paragraphs describing how many, which and the shape of the samples for the various tests are confusing. I suggest to make it clearer and insert a table

Results

It is essential to better explain the link between toughness and the static tests carried out.

Comments on the Quality of English Language

Minor editing of English language required

Author Response

The paper “Betholletia excelsa fruit: unveiling toughening mechanisms and biomimetic potential for advanced materials” is interesting and centered on the scope of the journal. The paper is suitable to publish but some modifications are required.

 Introduction

The introduction is well written, but I suggest to add a paragraph about to further explore the possible applications. It is not enough just to say that the work may be useful for composites filled with natural fibers.

We believe that comprehending the properties of the brazil nut mesocarp can yield valuable insights for structural composites, such as glass and carbon fiber-reinforced composites and not only natural fiber composites.

We added the following explanation.

“The Bertholletia excelsa mesocarp shows great resemblance to some fiber reinforcements composites. For instance, it has fibers in different orientations forming a sandwich structure while involved by a foam-like matrix with very different properties. Therefore, the understanding of how this biological material achieves outstanding mechanical performance can provide useful insights to optimize mechanical properties while reducing density of fiber reinforced composites. Such biological capsule can inspire improvements on pipes/tubes, pressure vessels, tanks,  masts, missile cases, rocket motor cases, aircraft fuselages and protective components for commodities or persons.”

Materials and methods

The paragraphs describing how many, which and the shape of the samples for the various tests are confusing. I suggest to make it clearer and insert a table

A table was inserted in the supplementary information.

Results

It is essential to better explain the link between toughness and the static tests carried out.

In fact, we did not directly measure the toughness of the mesocarp in the tests we conducted. The link between toughness and the tests carried out is the toughening mechanisms observed. Therefore, we have demonstrated and extensively discussed how the microstructure of the mesocarp deflects and branches advancing cracks, resulting in a tortuous crack path in the static tests. Additionally, we cite several well-known toughening mechanisms, as fiber bridging, fiber pullout, plastic deformation. As mentioned in the Discussion, these features are highly desirable in materials intended for high fracture toughness.

Round 2

Reviewer 2 Report

Comments and Suggestions for Authors

Thank you very much for improving the manuscript.

It is now an interesting work.

Comments on the Quality of English Language

Some minor spelling or grammatical errors.